# Review on Molecular Dynamics Simulations of Effects of Carbon Nanotubes (CNTs) on Electrical and Thermal Conductivities of CNT-Modified Polymeric Composites

**Lida Najmi and Zhong Hu** *

Department of Mechanical Engineering, South Dakota State University, Brookings, SD 57007, USA
* Correspondence: zhong.hu@sdstate.edu; Tel.: +1-605-688-4817

**Abstract:** Due to the unique properties of carbon nanotubes (CNTs), the electrical and thermal conductivity of CNT-modified polymeric composites (CNTMPCs) can be manipulated and depend on several factors. There are many factors that affect the thermal and electrical conductivity of CNTs and CNTMPCs, such as chirality, length, type of CNTs, fabrication, surface treatment, matrix and interfacial interaction between the matrix and reinforcement (CNTs). This paper reviews the research on molecular dynamics (MD) simulations of the effects of some factors affecting the thermal and electrical conductivity of CNTs and CNTMPCs. First, the chirality dependence of the thermal and electrical conductivity of single-walled carbon nanotubes (SWNTs) was analyzed. The effect of chirality on the conductivity of short-length CNTs is greater than that of long-length CNTs, and the larger the chiral angle, the greater the conductivity of the CNTs. Furthermore, the thermal and electrical conductivity of the zigzag CNTs is smaller than that of the armchair one. Therefore, as the tube aspect ratio becomes longer and conductivity increases, while the effect of chirality on the conductivity decreases. In addition, hydrogen bonding affects the electrical and thermal conductivity of the CNTMPCs. The modeling of SWNTs shows that the thermal and electrical conductivity increases significantly with increasing overlap length. MD simulations can be effectively used to design highly conductive CNTMPCs with appropriated thermal and electrical properties. Since there are too many factors affecting the thermal and electrical conductivity of CNTMPCs, this paper only reviews the effects of limited factors on the thermal and electrical conductivity of CNTs and CNTMPCs based on MD simulations, and further detailed studies are required.

**Keywords:** carbon nanotubes (CNTs); hydrogen bonding; CNT-modified polymeric composites (CNTMPCs); chirality; aspect ratio; molecular dynamics (MD) simulation

## 1. Introduction

The need of human communities and industries to build lightweight, high-strength, durable structures and highly conductive materials has increased the demand for polymeric composites. Carbon nanotube (CNT)-modified polymeric composites have specific characterizations that can improve the properties of materials. Materials with high electrical and thermal conductivity are useful for many applications, such as in general industry, aerospace and high-tech industries, and even everyday items. Improper or uncontrolled heat transfer is a widespread concern that affects the performance, reliability, and lifetime of materials, especially electronic devices [1]. High electric conductivity is important for the manufacture of various materials, such as electrical machines, starters, and rheostats [2]. The thermal and electrical conductivity of nano materials play a crucial role in controlling the performance and stability of nanocomposites such as nano/micro devices. CNTs have special properties such as high strength, lightweight, unique electronic structure, and high stability, making them ideal materials with a wide range of applications. CNTs can be categorized as single-walled CNTs (SWNTs), double-walled CNTs (DWNTs), and multi-walled

CNTs (MWNTs) according to the number of layers. SWNTs are a sheet of graphite (a hexagonal lattice of carbons) rolled into a cylinder and MWNTs have two structural models [3]. In the Russian Doll model, the CNT contains another nanotube inside (the inner nanotube has a smaller diameter than the outer nanotube). In the Parchment model, a single graphene sheet rolls around itself multiple times, resembling a rolled paper scroll [4]. MWNTs have similar properties to SWNTs, but the outer walls of MWNTs can protect the inner CNT from chemical interactions with external materials. CNTs have unique conductive properties and depend on their structure, functional bonding, and synthesis method, their thermal conductivity varies significantly from 660 W/m·K [5] for individual SWCNTs to the values below 0.1 W/m·K indicating thermal insulators for MWNT bundle system [6,7]. They can be either metallic or semiconducting, depending on their chirality. Depending on how the hexagons are arranged along the tube axis, SWNTs can form three different designs: armchair, chiral and zigzag. An armchair CNT has electrical properties similar to metals, but the other two structures have electrical properties similar to semiconductors [8]. Using CNTs with a higher aspect ratio (length/diameter) is an efficient means to obtain better thermal conductivity enhancement for CNT-modified polymer composites. Thus, as the tube length increases, the thermal conductivity increases, while the effect of chirality on the thermal conductivity decreases [9].

Another factor affecting the conductivity of CNT-modified composites is the length of the CNTs, or the aspect ratio of the CNTs. Bonding two CNTs together can make the pristine CNTs longer and serves as a pathway for electrons or phonon flow (heat transfer). For this reason, hydrogen bonds play a very important role in conducting heat and electricity. Instead of forming covalent bonds with hydrogen atoms, CNTs can form hydrogen bonds with special dipole–dipole attraction between molecules. It is caused by the attractive force between a covalently bonded atom, such as an N, O or F atom, and another very electronegative atom [10]. In fact, the large difference in electronegativity between H atoms and N, O or F atoms results in highly polar covalent bonds. Due to the difference in electronegativity, the H atom has a larger partial positive charge, while the N, O or F atom has a larger partial negative charge.

Due to the multi-factors affecting the performance of the CNT-modified composites, it is not yet fully quantitative to understand how these factors affect the material performance from the nanoscale to the macroscale, which is required to effectively design and develop novel composites with desired properties for applications. For this reason, the traditional trial-and-error experimental approaches are very expensive and time-consuming. Depending on the problem and spatial and temporal scales of interest, various approaches to materials design based on computer modeling are advancing, ranging from quantum mechanics to continuum simulations. Molecular dynamics (MD) or first principles simulations are ideal for studying nanoscale material properties. MD is an atomistic scale simulation that describes the interactions between atoms through interatomic potentials. In the MD method, electronic effects are averaged, and the time evolution of atomic positions and velocities are calculated according to Newton's equations of motion. The electron-dependent approximation is based on the Born–Oppenheimer theory, and the MD time step used to describe atomic motion is sufficient for electrons to achieve their ground stable states, compared to nuclei due to mass differences. The interatomic potentials (force fields) are developed from the first principles or experimentally to describe the interactions between the atoms, including the effect of electrons, in terms of reproducible forces. The reliability of the interatomic potentials determines the accuracy of the MD simulations and is also related to the ability to bridge the effectiveness of mesoscale methods [11–17]. The polymer matrices and CNTs can interact through strong covalent or electrostatic interactions or hydrogen bonding. These chemical interactions lead to strong coupling at the interface. Alternatively, the polymer matrices can interact with CNTs through weak electrostatic interactions, such as van der Waals forces [18–22]. These detailed considerations are very important when considering the design and optimization of the load/electron/phonon transferring/passing through the interface.

Therefore, in this paper, the effects of CNTs on the electrical and thermal conductivity of CNT-modified polymer composites based on MD modeling were reviewed, and the data were analyzed and discussed, which will help on how to increase the thermal or electrical conductivity of CNT-modified polymer composites and help the material design and development for thermal and electrical conductive applications.

## 2. Fundamental Concepts

### 2.1. Dispersion of Carbon Nanotubes

CNTs have specific properties. To achieve the desired properties, CNTs should be well dispersed in solvents or polymer solutions, whereas the controlled dispersion of CNTs in solutions or composite matrices remains a challenge due to the strong van der Waals binding energy aggregates associated with the CNTs [23,24]. For proper dispersion, a two-step process should be taken. The first one is a mixing/sonication process and the second one is a stabilization process [25]. The mixing process can be described as the delivery of mechanical energy into a solution to separate aggregates. This mechanical energy generates localized shear stress that eventually leads to dispersion through the rotation of mixer blades or cavitation in sonication. This supplied energy should be lower than that required to fracture the nanotubes. After removing the external shear stress, the CNTs in solution reconfigure themselves into a new low-energy equilibrium state by re-agglomeration. To prevent re-agglomeration, surfactants are added to provide steric hindrance or electrostatic charge repulsion to stabilize the particles [26]. The driving force for re-agglomeration should be greater than the van der Waals attraction [27]. Sodium dodecyl sulfate (SDS) is one of the most used surfactants.

Another non-covalent approach is polymer wrapping. The suspension of CNTs in the presence of a polymer, such as poly (phenylene vinylene) or polystyrene, lead to the polymer wrapping around the CNTs to form a super-molecular complex of CNTs. The polymer wrapping process is achieved through van der Waals interactions and π–π stacking between CNTs and polymer chains containing aromatic rings. Furthermore, the endohedral method is another non-covalent method for CNT functionalization. In this method, guest atoms or molecules are stored in the inner cavity of CNTs through the capillary effect. Insertion usually occurs at defects located on the ends or sidewalls. The insertion of inorganic nanoparticles into the tubes, such as C60, Ag, Au and Pt, and small biomolecules such as proteins and DNA are typical examples of endohedral functionalization [26,28].

### 2.2. Thermal Conductivity

Heat transfer is the movement of thermal energy from a warmer region to a cooler region. To understand the thermal conductivity of materials, it is important to be familiar with the concept of heat transfer, which occurs in several circumstances: (a) when there is a temperature gradient within an object, (b) when an object is at a different temperature when it is in contact with another and (c) when an object is at a temperature different from its surroundings [29].

The direction of heat transfer is determined by the second law of thermodynamics, which means that heat transfer is always from a warmer region to a cooler region and continues until thermal equilibrium is reached [30].

Thermal conductivity, *k*, is a material property that represents the ability to conduct heat. The heat flux given by Fourier's first law is proportional to the temperature difference, surface area and length of the heat transfer in the sample [31]:

$$H = \frac{\Delta Q}{\Delta t} = kA \frac{\Delta T}{\Delta l} \tag{1}$$

where A is the surface area, $\Delta l$ is the heat transfer length, and $\Delta Q/\Delta t$ is the heat transfer rate. Therefore, as the surface area and temperature difference increase, and the length decreases, the thermal conductivity will increase [32].

### 2.3. Electrical Conductivity

The electrical conductivity of a material is defined as the amount of electric charge transferred on a unit area per unit time under the action of a unit potential gradient:

$$J = \sigma E \tag{2}$$

where $E$ is the potential gradient and $J$ is the current density (current per unit area). The electrical conductivity of an isotropic material is $\sigma = \frac{1}{\rho}$ and electric resistivity, $\rho$, for a real sample of length $l$, cross-sectional area A, and resistance $R$ is calculated by

$$R = \rho \frac{l}{A} \tag{3}$$

Therefore, as the surface increases, the electrical conductivity will increase [33].

### 2.4. Chirality of CNTs

The vector $\boldsymbol{R} = m\,\boldsymbol{a_1} + n\,\boldsymbol{a_2}$ represents the chirality and diameter of SWNTs, where $\boldsymbol{R}$ is the lattice vector of two-dimensional graphene, $m$ and $n$ are integers; $\boldsymbol{a_1}$ and $\boldsymbol{a_2}$ are the unit vectors of the graphene. The schematic diagram of the chirality of a graphene is shown in Figure 1, and the diameter of a SWNT is defined as

$$d = \frac{|R|}{\pi} = a\,\frac{\sqrt{m^2 + mn + n^2}}{\pi} \tag{4}$$

where $a = 1.42\sqrt{3}$ (nm) is the lattice constant. When $m = n$, the SWNT is called an armchair SWNT. When $m = 0$ or $n = 0$, the SWNT is named a zigzag SWNT. Chirality includes crossing angles ranging from zigzag SWNTs to armchair SWNTs with a crossing angle of $30°$ in between. Therefore, the chiral angle is defined as the angle between the vector $\boldsymbol{R}$ and the orientation of the zigzag SWNT [34,35]. It is represented as

$$\theta = \tan^{-1}\left|\sqrt{3}n/(2m+n)\right| \tag{5}$$

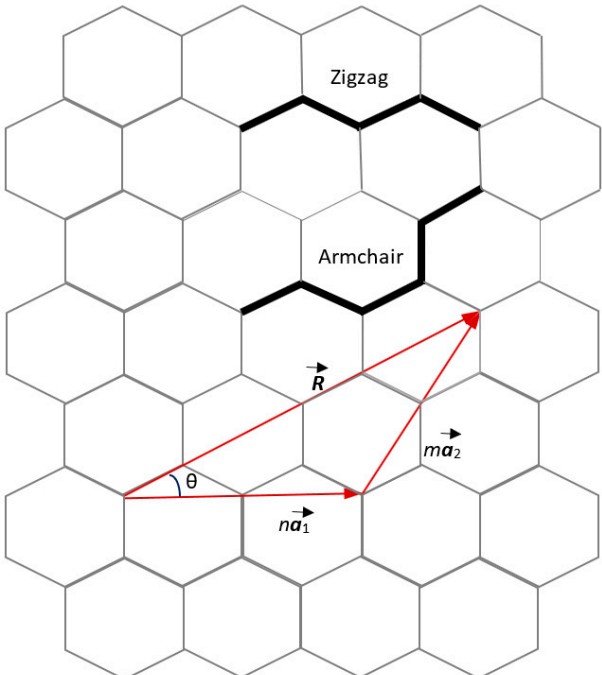

**Figure 1.** Schematic diagram of the chirality of a graphene hexagonal lattice with lattice vectors $\boldsymbol{a_1}$ and $\boldsymbol{a_2}$.

### 2.5. Some of the MD Techniques Used in the Literatures of This Review

A schematic diagram of the structure of a SWNT and a temperature profile generated along the axial length of the SWNT used in the MD simulations is shown in Figure 2. A nonequilibrium molecular dynamics (NEMD) was employed to calculate the thermal conductivity of SWNTs. To prevent sublimation throughout the simulation, the ends of the SWNT were fixed as frozen walls in which the atoms remained stationary. Next to the two walls, the blue area is the cold region and the red area is the hot region [9,36,37].

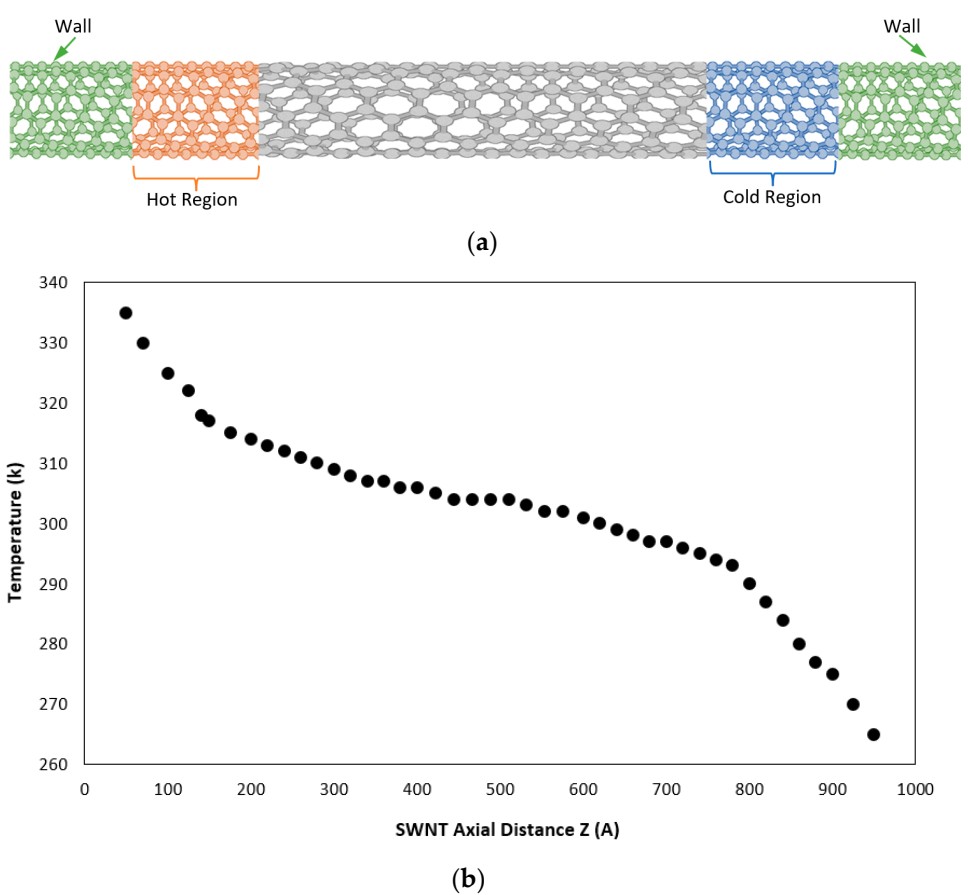

**Figure 2.** (**a**) Schematic structure of a SWNT used in MD simulations and (**b**) a typical temperature profile along the axial length of a SWNT with a chirality (12, 12).

By adding heat flux to the system, a non-equilibrium conductance is established, resulting in a steady temperature gradient, and the thermal conductivity can be obtained by Fourier's law [38]

$$\lambda = -\frac{J}{\partial T / \partial z} \tag{6}$$

where $\partial T / \partial z$ is the temperature gradient along the z-axis and $J$ is the heat flux [39].

One of the atomic potential functions used in the MD simulations is the Adaptive Intermolecular Reactive Empirical Bond Order (AIREBO) [40] potential to describe carbon–carbon interatomic interactions. AIREBO has been widely used to study heat transfer problems in carbon materials [41,42]. It has the following form:

$$E = \frac{1}{2} \sum_i \sum_{j \neq i} [E_{ij}^{REBO} + E_{ij}^{LJ} + \sum_{k \neq i,j} \sum_{l \neq i,j,k} E_{kijl}^{TORTION}] \tag{7}$$

where the hydrocarbon $E^{REBO}$ potential was developed by Brenner. Here, the $E^{REBO}$ used to describe the interactions between covalent atoms has the same form as in [43,44] and has the same coefficients as the Brenner potential function, that is:

$$E_{ij}^{REBO} = V_{ij}^{R} + b_{ij}V_{ij}^{A} \tag{8}$$

where $V_{ij}^{A}$ and $V_{ij}^{R}$ are the attractive and repulsive interaction energies between atoms *i* and *j*, respectively, and the many-body bonding parameters are notated by $b_{ij}$.

The long-range interactions between non-bonded atoms are based on the Lennard–Jones 12-6 potential, represented by the $E^{LJ}$ term, defined as [45]:

$$E_{ij}^{LJ} = 4\varepsilon_{ij}[(\frac{\sigma_{ij}}{r_{ij}})^{12} - (\frac{\sigma_{ij}}{r_{ij}})^{6}] \tag{9}$$

Dihedral-angle intermolecular interactions, the role of which is considered unimportant in CNT analysis [40], are described by the $E^{TORTION}$ term.

The cross-sectional area of a SWNT is a ring $S = \pi\delta D$. Among them, $\delta$ is the wall-thickness that uses a van der Waals thickness of 0.34 nm, D is the diameter of the SWNT, and the bond length of the carbon–carbon in the simulations is 0.144 nm. [46]. In the NEMD simulation, in order to establish a stable temperature gradient, a stable heat flux is added to the system. A certain amount of kinetic energy is taken from the cold region, and at the same time, the same amount of kinetic energy is added to the hot region. These processes are achieved using the variants of the algorithm proposed by Jund and Jullien [47]. By rescaling the velocity of atoms, the heat flux is added to or taken away from the corresponding region.

All MD simulations can be conducted using the Large-scale Atomic/Molecular Massively Parallel Simulator (LAMMPS) [48], one of the powerful open-source MD simulation packages, with a time step as small as $\Delta t = 0.1$ fs. First, a 100 ps relaxation time for the velocity and the constant volume without thermostat (NVE) will be provided to relax the entire system at a room temperature of 298 K. A stable temperature gradient will be reached throughout the system. After the equilibration, a constant heat flux will be applied to the system for another 400 ps. Along the axial direction (the heat flow direction), the CNTs with the same thickness will be separated into slabs. Through the kinetic energy of the atoms in the slabs, the local temperature of each slab can be obtained. The last 0.1 ns will be averaged to obtain a smooth temperature profile. Figure 2b shows the temperature profile of (12, 12) SWNT along the length of the tube. It is about 90 nm long between the cold region and the hot region. The thermal conductivity of the SWNT can be calculated by fitting the temperature configuration within the middle linear part. Therefore, the temperature gradient can be easily obtained by calculating the slope of the temperature fitting line.

To study the electrical conductivity, under the periodic boundary conditions of $p = 1$ atm, the Nanoscale Molecular Dynamics (NAMD) simulation package can be used in the constant temperature and constant pressure (NpT) ensemble (an isothermal-isobaric ensemble), while T = 300 K, the time step is $\Delta t = 1$ fs, and the simulation time can be as long as 4 ns [41]. The Boltzmann transport theory implemented in CRYSTAL17 [42] can be used to obtain the electrical conductivity σ, at temperature T of 300 K. The constant relaxation time, right boundary and transport coefficient can be solved approximately, such that

$$[\sigma]_{qr}(\mu, T) = e^{2} \int dE \left(-\frac{df_{0}(\mu, T)}{df}\right) E_{qr}(E) \tag{10}$$

where $E$ is the energy of the system, $f_{0}$ is the Fermi–Dirac distribution, $\mu$ is the Fermi level, and $E_{qr}(E)$ is the transport coefficients defined as

$$E_{qr}(E) = \tau \sum_{k} \frac{1}{N_{k}} \frac{1}{V} \sum_{ij} v_{i,q}(k)v_{j,r}(k)\delta(E - E_{i}(k)) \tag{11}$$

where $V_{i,q}$ is the velocity of the *i*th band along q, $N_k$ is the number of *k*-points over the cell volume V, and the lifetime, $\tau$, is 1 fs for all studied systems. The conductivity is shown as the summation of the $\alpha$ and $\beta$ spin states [49,50].

### 2.6. Effectivity of Hydrogen Bonding for CNT Alignment

In the experimental method, carbon nanofillers are usually those materials with an average fiber diameter of 150 nm and lengths of 50–200 μm, industrial-grade hydroxyl functionalized multiwalled carbon nanotubes (MWNT-OH) with diameters of 20–40 nm and lengths of 10–30 μm, 88+% (>90%) purity, using Varathane's water-based polyurethane, with A + B epoxy and Polycrylic coating. The carbon nanomaterials are mixed into the base coating solutions with the desired weight percentage, and homogeneous mixture can be obtained using both a three-roll mill and a Branson Sonifier [51,52]. According to the thickness of the mixture, the resulting coating mixture can be applied to the non-sticky and non-conductive surface using a smear-casting technique or an airbrush, which is easy to release. Generally, in the experimental method, the electrical conductivity is calculated based on the inverse of the resistivity. The resistivity is given [51].

$$\rho = R\frac{A}{L} \tag{12}$$

where $R$ is the electrical resistance in ohms, $L$ is the length of the material in centimeters, and $A$ is the cross-sectional area in square centimeters. The electrical conductivity is simply the inverse of resistivity in units of Siemens per centimeter (S cm$^{-1}$), where S is 1/ohm.

The NEMD method [53] can be used to study the thermal conductivity for aligned CNT junctions. Figure 3a shows a schematic atomic modeling configuration of two parallel-aligned 10 nm long CNTs with (10, 10) chirality and an overlap length of 3.934 nm ($\Delta x$) by hydrogen bonds, where the red area is a hot region, the blue area is a cold region, and the heat flux is $q$. The thermal energy added to the hot region is equal to the thermal energy subtracted from the cold region. In addition, in order to prevent CNT rotation, as shown the grey region in Figure 3a, 40 carbon atoms at both ends of the two CNTs were fixed. The velocities of the carbon atoms in all directions were set to 0. In order to obtain the temperature profile of the system, the remaining region of each CNT was laterally divided into 10 slabs. The temperature of each slab was calculated as follows:

$$T = \frac{1}{3nk_B}\sum_{k=1}^{n} m_k v_k^2 \tag{13}$$

where n is the number of atoms in each slab, $k_B$ is the Boltzmann constant, and $m_k$ and $v_k$ are the atomic mass and velocity of the atom k, respectively [54].

The AIREBO potential and the Lennard Jones 12-6 potential were calculated as described in Equations (6) and (7). Shi et al. [55] investigated the thermal transport in 3D CNT-graphene structures by MD simulations and found bonding interactions between CNT-graphene junctions. The system was initially equilibrated at a temperature of 300 K for 200 ps with an integration step of 0.2 fs using a Nose–Hoover thermostat [56], then a consecutive micro-canonical ensemble (NVE) was adopted for another 200 ps. Finally, a constant heat flux q was applied to the system for additional 1.2 ns to reach a steady state. The temperature was collected during the last 400 ps period, and the temperature profile is shown in Figure 3b. it is important for thermal conductance calculation. The moderate temperature drop at the junction is important, because a larger temperature drop can cause a non-linear effect, resulting in statistical uncertainties due to temperature fluctuation [57]. In order to study the effect of overlap length on the thermal conductivity, heat fluxes ranging from 80 nW to 560 nW were applied across the overlap length from 0.982 nm to 6.886 nm to obtain a junction temperature drop of about ~50 K. Therefore, the effect of $\Delta x$ can be explained [58,59].

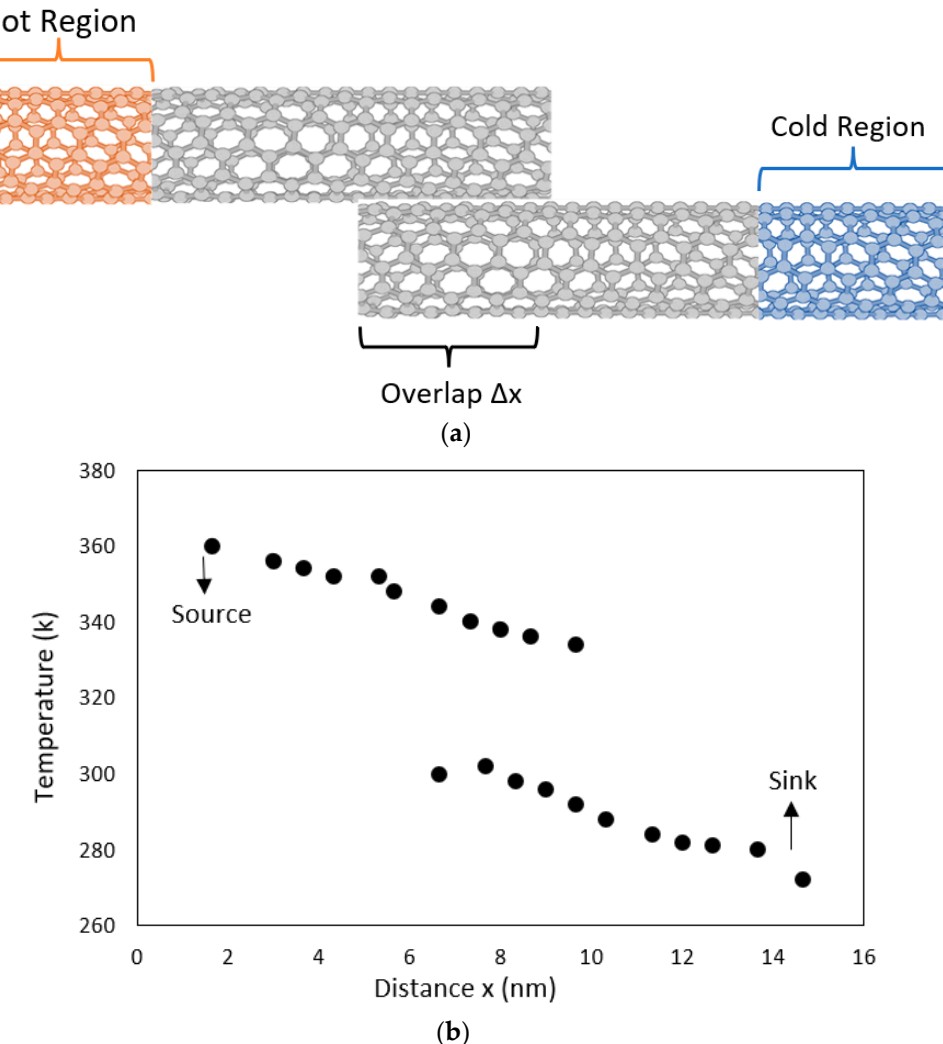

**Figure 3.** (**a**) The schematic diagram of the overlap of two parallel aligned CNTs by hydrogen bonds with an overlap length of 3.934 nm (Δx) and free boundary conditions in all directions. (**b**) The temperature profile of the corresponding structure.

For the electrical conductivity simulation, the selection of hydroxyl functionalized MWNTs can be used for the modeling system. The primary CNTs of chirality (7, 7), and a length of 5 nm. The same software package and settings as forementioned can be used to study the electrical conductivity, and the transport coefficients described in Equations (8) and (9) [60,61] can be calculated.

## 3. Results and Discussion

### 3.1. Effects of CNT Chirality and Length

The simulation results were gathered from references [62–67] and the simulation results were grouped in terms of CNTs' geometry, and the simulation results into three groups listed in Table 1, compared and analyzed: (a) Group A is SWNTs with a smaller diameter of 1.66 nm and a tube length of 20 nm; (b) Group B is SWNTs with a larger diameter, which is twice as large as the smaller diameter ones, and a tube length of 20 nm; and (c) Group C is SWNTs with a smaller diameter of 1.66 nm, the same diameter as that in Group A, and a tube length of 50 nm. The thermal conductivity results also plot in Figure 4, where the horizontal error bars represent the data deviation of the CNTs' chiral angle, and the vertical error bars represent the error deviation of the thermal conductivity data measured.

**Table 1.** Chiral angle-dependent thermal conductivities of group A, B and C of SWNTs [46,62–68].

| Group A | | Group B | | Group C | |
|---|---|---|---|---|---|
| Chiral Angle (°) | Thermal Conductivity (W/m·K) | Chiral Angle (°) | Thermal Conductivity (W/m·K) | Chiral Angle (°) | Thermal Conductivity (W/m·K) |
| 5.0 | 105.6 | 1.0 | 110.3 | 5.0 | 125.2 |
| | | 5.0 | 100.2 | | |
| 12.5 | 112.5 | 7.5 | 118.1 | 13.0 | 130.2 |
| | | 12.5 | 130.4 | | |
| 17.5 | 134.2 | 15.0 | 140.2 | 17.5 | 122.2 |
| | | 20.0 | 150.1 | | |
| 22.5 | 130.3 | 22.5 | 125.1 | 23.0 | 132.1 |
| | | 25.0 | 140.3 | | |
| 27.5 | 110.1 | 27.5 | 120.0 | 28.0 | 139.1 |
| | | 30.0 | 122.3 | | |

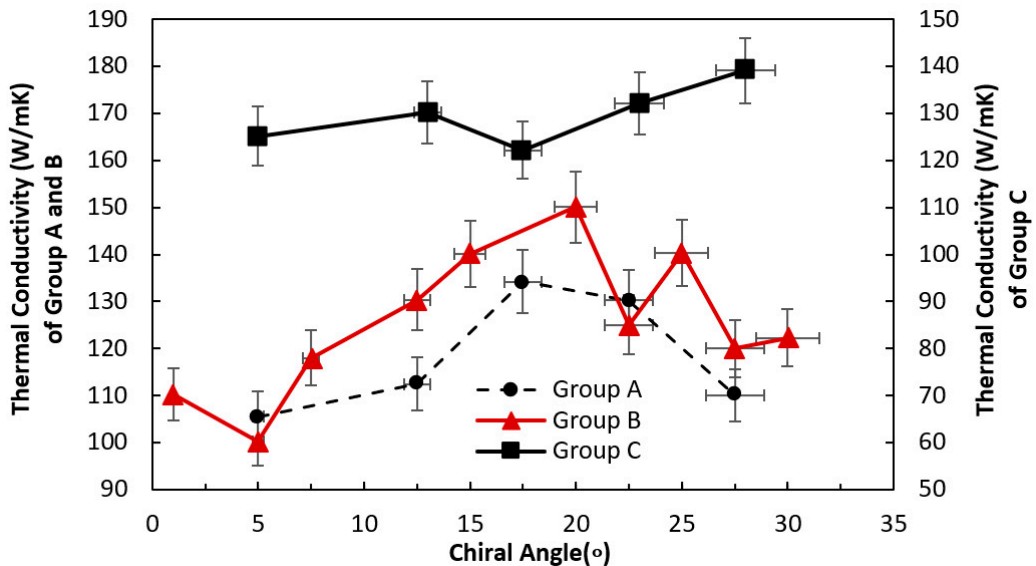

**Figure 4.** Chiral angle vs. thermal conductivity of smaller diameter shorter SWNTs with a tube length of 20 nm (Group A), larger diameter shorter SWNTs with a tube length of 20 nm (Group B), and smaller diameter longer SWNTs with a tube length of 50 nm (Group C).

For Group A, the thermal conductivity of chiral SWNTs increases at first until reaches the peak value of 134.2 (W/m K) at a chiral angle of 17.5°, and then decreases with the increase in the chiral angle. A chiral angle of 5.0° shows a lowest thermal conductivity of 105.6 (W/m·K), while a chiral angle of 17.5° shows a highest thermal conductivity of 134.2 (W/m·K). The difference between the maximum and the minimum values is about 28.6 (W/m·K) [62,63].

In order to observe the effect of CNTs' diameter, Group B was introduced [62,64]. It includes SWNTs with the smaller diameter of Group A, i.e., 1.66 nm, as well as zigzag and armchair SWNTs. As shown in Figure 4, the thermal conductivity for Group B is similar to that of Group A, but more complicated. The peak value of 150.1 (W/m·K) happens at 20°, and the difference between the maximum and minimum values is 49.9 (W/m·K). The effect of chirality becomes more pronounced. The data from Group A and Group B in Figure 4 and Table 1 provide more details of the effect of chirality on thermal conductivity. However, the effect of the chirality is not a monotonically changing pattern (monotonically increasing

or monotonically decreasing); instead, there is an inflection point (threshold value) where the CNTs' thermal conductivity is at a maximum point that can be fully accounted for when we conduct effective new material design/development.

Since these chiral angles are between 0° and 30°, this reflects the thermal conductivity of the chiral SWNTs depending on their chiral angle.

Group C is an extended version of Group A with longer CNTs having the lowest and the highest thermal conductivity of 122.2 (W/m·K) and 139.1 (W/m·K), respectively [65–67], and the minimum value is at 17.5°. The thermal conductivity of Group C looks flatter; the difference between the maximum and minimum values is only about 16.9 (W/m·K). It can be concluded that chirality has less effect on those longer SWNTs.

Figure 5 and Table 2 list the electrical conductivities of the armchair SWNTs and the zigzag SWNTs with different volume fractions of CNTs in a composite. As we can see, for the armchair SWNTs with aspect ratios (the ratio of the length to the diameter) ranging from $7.5 \times 10^3$ to $15 \times 10^3$, the electrical conductivity is much higher than that of the zigzag SWNTs. For lower amounts of CNTs, the electrical conductivities of the armchair and the zigzag SWNTs have increased sharply with the increase in the CNT volume fraction, but this upward trend continues as the slop of higher volume fraction decreases. For a high-volume fraction of CNTs, the electrical conductivity starts to decrease, mainly because CNTs cannot be distributed well in the matrix of the composite material very well [66,68–70].

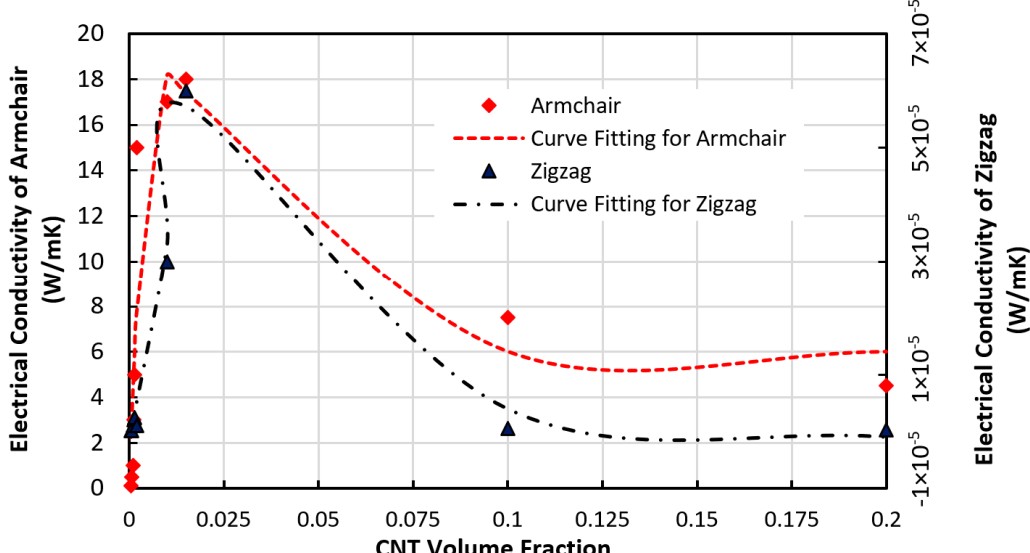

**Figure 5.** Electrical conductivities of armchair and zigzag SWNTs vs. CNT volume fraction in a composite with aspect ratios ranging from $7.5 \times 10^3$ to $15 \times 10^3$ [68,70].

The relationship between electrical conductivity and volume fraction of zigzag and armchair CNTs is similar. So, this relationship based on data, which is shown in Figure 5 and listed in Table 2, can be represented by a curve-fitting equation as:

$$\sigma_c = A(1 - e^{-BV_f}\left(\cos\left(CV_f\right) + D\sin\left(CV_f\right)\right)) \tag{14}$$

where A, B, C and D are 6.01, 104.84, 9.1 and −74.75 for armchair CNTs, and $1.8 \times 10^{-7}$, −37.12, 31.43, and −394.3 for zigzag CNTs, respectively. $\sigma_c$ represents electrical conductivity and $V_f$ stands for volume fraction of CNTs.

**Table 2.** The electrical conductivities of armchair and zigzag SWNTs with different volume fractions of CNTs in a composite with an aspect ratio ranging from $7.5 \times 10^3$ to $15 \times 10^3$, and with different lengths at a volume fraction of 0.0005 and a diameter of 1–2 nm [66,70].

| Length (mm) | Armchair Electrical Conductivity (m/s) | Zigzag Electrical Conductivity (m/s) |
|---|---|---|
| 0.01 | $2 \times 10^{-2}$ | $3 \times 10^{-8}$ |
| 0.015 | $10^{-1}$ | $10^{-7}$ |
| 0.6 | 3.02 | $2 \times 10^{-6}$ |
| 0.8 | 3.15 | $2.4 \times 10^{-6}$ |
| 0.9 | 3.52 | $2.8 \times 10^{-6}$ |
| 1.1 | 3.79 | $2.9 \times 10^{-6}$ |
| 1.3 | 3.81 | $3 \times 10^{-6}$ |
| Volume Fraction | Armchair Electrical Conductivity (m/s) | Zigzag Electrical Conductivity (m/s) |
| 0.0005 | $10^{-1}$ | $10^{-7}$ |
| 0.00075 | 0.5 | $2 \times 10^{-7}$ |
| 0.0010 | 1 | $10^{-6}$ |
| 0.0012 | 3 | $2 \times 10^{-6}$ |
| 0.0015 | 5 | $2.5 \times 10^{-6}$ |
| 0.0020 | 15 | $10^{-5}$ |
| 0.010 | 17 | $3 \times 10^{-5}$ |
| 0.015 | 18 | $6 \times 10^{-5}$ |
| 0.1 | 7.5 | $0.5 \times 10^{-6}$ |
| 0.2 | 4.5 | $0.25 \times 10^{-6}$ |

The length of the SWNTs affects the electrical conductivity. As shown in Figure 6 and Table 2, for CNTs with a diameter of 1–2 nm and a volume fraction of 0.0005, the electrical conductivities of the armchair and the zigzag SWNTs with different lengths are different, and the electrical conductivity increases with the increase in the length of SWNTs of both the armchair and zigzag SWNTs. Therefore, the longer CNTs can ensure a more effective electron conduction pathway along the individual nanotubes. From Figure 6, the electrical conductivity levels off for longer CNT lengths. Contact resistance at the interfaces of individual nanotubes plays a role. As the CNTs become longer, the overlapping of the nanoconductors creates more electrical connections, and the presence of a large number of junctions in the fibers increases the contact resistance. This increase continues until the contact resistance exceeds the conductivity. Therefore, the electrical conductivity of CNTs tends to be flat for longer CNTs. The relationship between the electrical conductivity and the length for zigzag and armchair CNTs based on the data shown in Figure 6 can be represented by the curve-fitting equation as:

$$\sigma_c = A\left(1 - e^{-BL}\right) \tag{15}$$

where *A* and *B* are 4.13 and 2.05 for armchair CNTs, and 3.73 and 1.33 for zigzag CNTs, respectively. $\sigma_c$ represents electrical conductivity and *L* stands for length of CNTs.

According to the data listed in Table 2, the electrical conductivity of the armchair CNTs is much higher than that of the zigzag CNTs. The chirality of a SWNT uniquely determines its atomic geometry and electrical structure. Studies have shown that the

structured armchair CNTs behave like metallic conductors, and the structured zigzag CNTs behave like semiconductors [68,71,72].

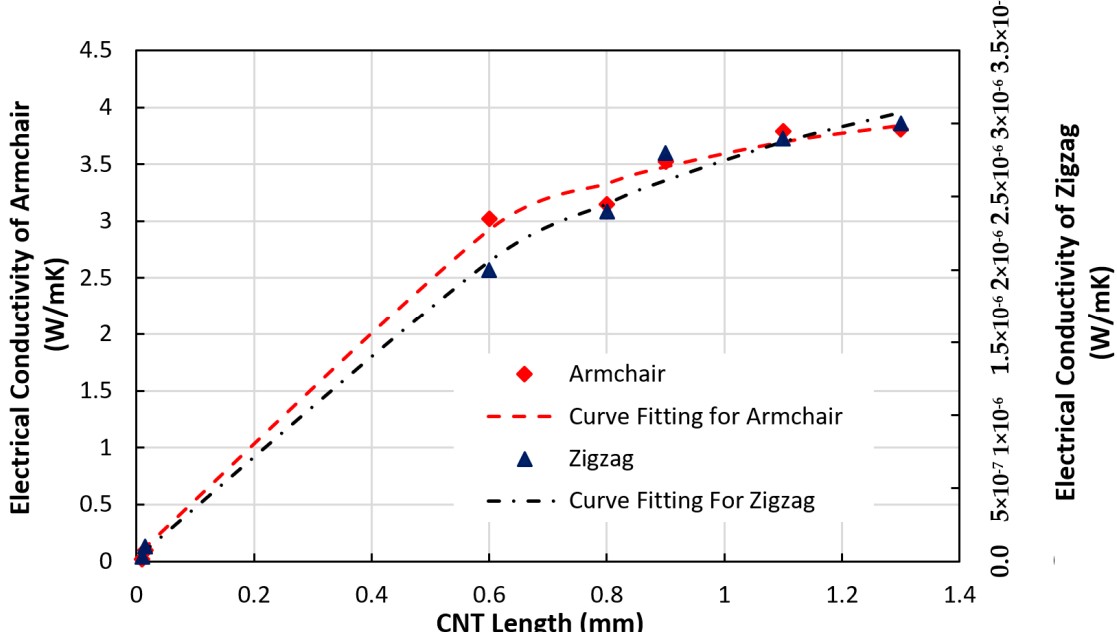

**Figure 6.** Electrical conductivities of armchair and zigzag SWNTs in a composite vs. CNT length at a CNT volume fraction of 0.0005 and a diameter of 1–2 nm [68,72].

### 3.2. Effect of CNT Overlap Length by Hydrogen Bonds

Zhang et al. conducted NEMD simulation and found that the thermal conductivity of the hydrogen bonded crystal polymer nanofibers may be 1 or 2 orders of magnitude higher than that of the typical engineering polymers [73]. The study by Lou et al. demonstrated the ability to dramatically lower the electrical resistivity of modified composites by adding a low weight percentage of CNTs and introducing hydrogen bonds [51]. Using hydroxyl-functionalized CNTs (f-CNTs) and organic solvents, the hydrogen bonds formed between nanotubes themselves or between the nanotubes and liquid molecules are attributed to the increase in conductivity [74].

Lou et al. [10] used industrial-grade MWNT-OH, which can form hydrogen bonds because CNTs are functionalized with hydroxyl groups. By adding appropriate solvents, these hydrogen bonds may form between the tubes and the base fluid [75]. Since the nanotubes may form weak networks, the hydrogen bonds can provide better connections and narrow the gaps between them. In general, the properties of CNTs in bulk materials allow better use of the hydrogen bonds [76–78]. The results show that the main reason for the extraordinary electrical conductivity is the hydrogen bonding formed between CNTs or between CNTs and water-based coating formulations [79,80]. The data show that the resistivities of 75% polycrylic/25%$H_2O$ and 75% polyurethane/25% $H_2O$ containing 4.5 wt.% MWNT-OH are 80 and 4.1 $\Omega \cdot cm$, respectively, 8 times and 207 times lower than the resistivities of the samples without $H_2O$, respectively. Therefore, the composites of 75% polycrylic/25%$H_2O$ and 75% polyurethane/ 25% $H_2O$ with 4.5 wt.% MWNT-OH have a higher electrical conductivity than that of the samples without $H_2O$. Table 3 shows the effect of $H_2O$ on the electrical conductivity and resistivity of CNT-modified composites [51,81,82].

Similarly, the f-CNTs have the same effect on thermal conductivity. By adding 4.5 wt.% MWNT-OH to the polycrylic matrix samples, the thermal conductivity increases by 43.4%, while the 75 wt.% polycrylic/25 wt.% $H_2O$ with the same MWNT-OH concentration, the thermal conductivity increases by 82.6%.

However, the data show that $H_2O$ does not always help increase the thermal conductivity of the f-CNT reinforced samples. When adding 25 wt.% $H_2O$ to a polyurethane

sample with 4.5 wt.% MWNT-OH, the thermal conductivity did not increase significantly. The reason is that as the concentration of carbon nanofillers increases, $H_2O$ acts more like hydrogen bonding auxiliary than a dispersing aid, while extra hydrogen bonds are conducive to electrical conductivity rather than thermal conductivity [83,84]. Therefore, as the gaps between the tubes and the fibers decrease, the thermal conductivity will not increase, and physical contact needs to improve the thermal energy transport.

Figure 7 shows that G increases with the increase in overlap length, where G is the thermal conductance and $\Delta x$ is the CNT overlap length. When $\Delta x$ increases from 0.982 to 6.877 nm, G increases from 1.00 to 11.76 nW/K. When the overlap length is greater, more carbon atoms of the two CNTs at the connection are involved and promote thermal transport across the junction. Therefore, this directly enhances the thermal conductance G [85–87]. The thermal conductance per unit overlap length is defined as $\sigma = G/\Delta x$, as shown in Figure 7, where $\sigma$ is a function of $\Delta x$, and $\sigma$ increases with the overlap length $\Delta x$ [88]. In addition, it was found that the CNT length at the bonded junction affects the thermal conductivity of the intertubes. Yang et al. [89] observed that when the tube length ranges from 24.56 nm to 123'nm, the intertube thermal conductivity varied from $1.46 \times 10^{-8}$ WK$^{-1}$ to $1.64 \times 10^{-8}$ WK$^{-1}$. For f-CNTs, hydrogen bonds are formed on the junction, and the thermal conductivity mainly depends on the number of hydrogen bonds. Therefore, as the overlap length increases, the hydrogen bond increases and the thermal conductivity increases.

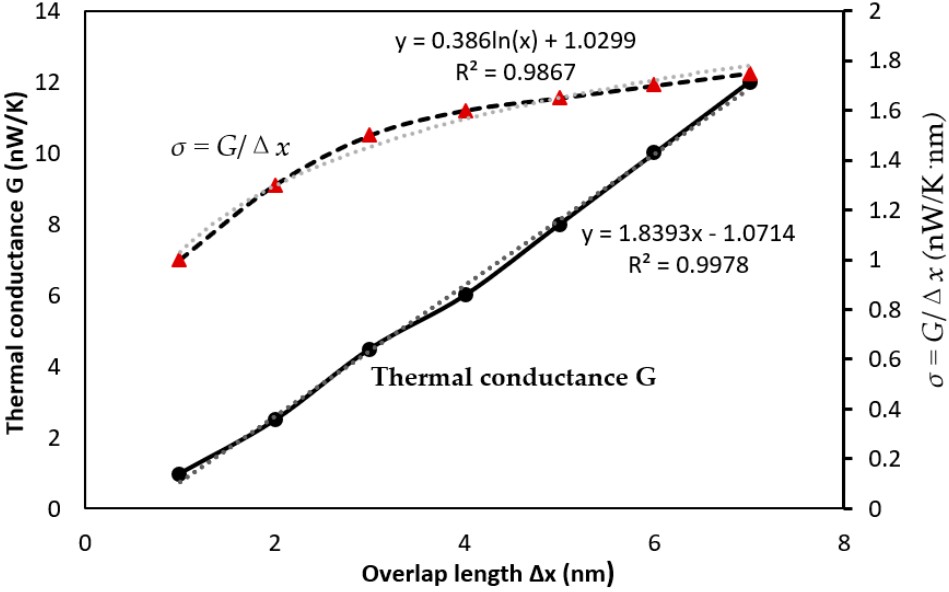

**Figure 7.** Thermal conductance G, and thermal conductance per unit overlap length $\sigma$ as a function of overlap length $\Delta x$.

**Table 3.** Measured electric resistivities and conductivities of polyurethane and polycrylic-based coatings containing 4.5%wt CNTs nanocomposites, with or without $H_2O$ [86].

| Base Coating | Resistivity ($\Omega$ cm) | Electrical Conductivity (S/cm) |
|:---:|:---:|:---:|
| Polyurethane | 850 | $1.2 \times 10^{-3}$ |
| 75% Polyurethane/25% $H_2O$ | 4.1 | 0.24 |
| Polycrylic | 690 | $1.45 \times 10^{-3}$ |
| 75% Polycrylic/25% $H_2O$ | 80 | 0.0125 |

### 3.3. Other Factors' Effects on Thermal and Electrical Properties of CNTs

Table 4 briefly summarizes the effects of chirality, length, and diameter on the thermal and electrical conductivity of CNTs, as reported by many other MD simulations. Many

reports have simulated thermal and electrical conduction of CNTs at 300 K. Variations in the nanotube length, diameter, chirality, boundary conditions and MD methods (EMD and NEMD) affect the range of simulated values.

**Table 4.** Summary of MD simulations of thermal conductivity values at room temperature for the maximum length and different diameters.

| Chirality | L (nm) | D (nm) | Thermal Conductivity (W/mK) | Electrical Conductivity (S/cm) | Simulation Method | Ref. |
|---|---|---|---|---|---|---|
| (10, 10) | <10 | 1.36 | 880 | 350 | EMD | [89] |
| (10, 10) | <1500 | 1.351 | 355 | 80 | NEMD | [90] |
| (10, 10) | 2.477–39.632 | 1.351 | 859 | 110 | | |
| (18, 0) | 2.145–34.320 | 1.404 | 790 | 95 | EMD | [91] |
| (14, 6) | 3.813–30.504 | 1.387 | 765 | 80 | | |
| (5, 5) | Aspect ratio | 0.68 | 4500 | 500 | | |
| (10, 10) | of 10–20 | 1.36 | 1700 | 300 | NEMD | [92] |
| (15, 5) | (~22 nm) | 1.41 | 1640 | 250 | | |
| (5, 5) | | 0.68 | 410 | 105 | | |
| (6, 6) | | 0.81 | 435 | 120 | | |
| (8, 8) | 12.2 and 24.4 | 1.08 | 365 | 103 | NEMD | [93] |
| (10, 10) | | 1.35 | 300 | 100 | | |
| (5, 5) | | 0.68 | ~1024 | 400 | | |
| (10, 10) | 6–100 | 1.36 | ~1023 | 380 | EMD | [94] |
| (15, 15) | | 2.03 | ~1022 | 365 | | |

## 4. Conclusions

This paper reviews the effects of the chirality, diameter and length of CNTs, as well as hydrogen bonding on the thermal and electrical conductivity of CNT-modified polymer composites based on MD simulations. The MD methods of EMD and NEMD simulation are briefly introduced, and the simulation results are compared and discussed.

The MD simulation results show that the chiral angle affects the thermal conductivity of SWNTs. The studies have shown that: (a) The thermal conductivity increases with the increase in the tube length, but the effect of chirality on the thermal conductivity decreases. (b) The thermal conductivity increases with the increase in the tube diameter, and the effect of chirality on the thermal conductivity increases but has little effect on the thermal conductivity. (c) SWNTs with larger chiral angles have greater thermal conductivity. Therefore, the thermal conductivity of the armchair SWNTs is higher than that of the zigzag SWNTs.

It has been observed that the chirality of CNTs affects their electrical conductivity, with the armchair SWNTs being more conductive than that of the zigzag SWNTs. Furthermore, the electrical conductivity of both armchair and zigzag structured SWNTs increases with increasing the length of SWNTs.

Studies have shown that f-CNTs can be achieved by hydrogen bonds due to the presence of $H_2O$, and with the increase in hydrogen bonds, the electrical conductivity increases and the resistivity decreases. However, for thermal conductivity, it depends on the composites; in some cases, the thermal conductivity increases in the presence of $H_2O$, but not in others. So, the extra hydrogen bonds are beneficial for electrical conductivity, but not always good for the thermal conductivity. In addition, the overlap length of the CNTs by hydrogen bonds affects the thermal conductivity, since the thermal conductivity also depends on the number of hydrogen bonds, and the thermal conductivity increases with the increase in the overlap length and the increase in the hydrogen bonds.

This article only outlines how some factors individually affect the electrical and thermal conductivity of CNTs and CNTMPCs. However, there are so many factors that play an important role in these materials' performance and development, such as chirality,

length, morphology, manipulation, volume fraction and surface treatment of the CNTs, solvent, matrix, environment (such as temperature, pressure and moisture), fabrication process and interfacial interaction between the matrix and the reinforcement, as well as the coupled effects of these factors. By fixing all possible factors as constants and allowing only one factor as an independent variable at a time, a great deal of further studies and simulations can be performance to better understand the mechanism by which each factor affects the material's performance.

**Author Contributions:** Conceptualization, Z.H. and L.N.; methodology, Z.H. and L.N.; formal analysis, L.N.; investigation, L.N. and Z.H.; resources, L.N.; data curation, L.N.; writing—original draft preparation, L.N.; writing—review and editing, Z.H.; visualization, L.N. and Z.H.; supervision, Z.H.; project administration, Z.H.; funding acquisition, Z.H. All authors have read and agreed to the published version of the manuscript.

**Funding:** This research received no external funding.

**Data Availability Statement:** Not applicable.

**Acknowledgments:** This work was supported by the J. J. Lohr College of Engineering and the Mechanical Engineering Department at South Dakota State University and are gratefully acknowledged.

**Conflicts of Interest:** The authors declare no conflict of interest. The funders had no role in the design of the study; in the collection, analyses, or interpretation of data; in the writing of the manuscript; or in the decision to publish the results.

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
