# Peer review of "Review on Molecular Dynamics Simulations of Effects of Carbon Nanotubes (CNTs) on Electrical and Thermal Conductivities of CNT-Modified Polymeric Composites"

_jcs, doi:10.3390/jcs7040165_

Round 1

Author Response

Please refer to the attachment (the cover letter to the editor with the responses to the reviewers. Thanks.

Reviewer 2 Report

Authors the effects of CNTs on the electrical and thermal conductivity of CNT-modified polymer composites based on MD modeling were reviewed. The review is well written and interesting to read. 

This version is suitable for publication in the journal

Author Response

(The authors gave the same response as above.)

Reviewer 3 Report

The review paper is timely and extensive. However, the following issues need to be addressed:

1, What is the novelty of this review? What are the main summaries?

2, The diagram is too simple; please modify it.

3. What is the outlook of this field?

Author Response

(The authors gave the same response as above.)

Round 2

Reviewer 1 Report

All of my concerns were well addressed, and the present version   is   acceptable for publication.                              

Reviewer 3 Report

The manuscript is in good shape now.